# Federated privacy-protected meta- and mega-omics data analysis in multi-center studies with a fully open-source analytic platform

Xavier Escriba-Montagut[1,2], Yannick Marcon[3], Augusto Anguita-Ruiz [1,2], Demetris Avraam[4], Jose Urquiza[1,2,5], Andrei S. Morgan [6,7], Rebecca C. Wilson [4], Paul Burton[8], Juan R. Gonzalez [1,2,5]*

1 Barcelona Institute for Global Health (ISGlobal), Barcelona, Spain, 2 Universitat Pompeu Fabra (UPF), Barcelona, Spain, 3 Epigeny, St Ouen, France, 4 Department of Public Health, Policy and Systems, University of Liverpool, Liverpool, United Kingdom, 5 Centro de Investigación Biomédica en Red en Epidemiología y Salud Pública (CIBERESP), Barcelona, Spain, 6 Université Paris Cité, Centre of Research in Epidemiology and StatisticS (CRESS), Obstetrical Perinatal and Pediatric Epidemiology Research Team (EPOPé), INSERM, INRAE, F-75006, Paris, France, 7 Elizabeth Garrett Anderson Institute for Women's Health London, University College London, London, United Kingdom, 8 Population Health Sciences Institute, Newcastle University, Newcastle, United Kingdom

* juanr.gonzalez@isglobal.org

**Data Availability Statement:** The source code used to recreate the use cases, as well as all the instructions and further clarifications on the code

## Abstract

The importance of maintaining data privacy and complying with regulatory requirements is highlighted especially when sharing omic data between different research centers. This challenge is even more pronounced in the scenario where a multi-center effort for collaborative omics studies is necessary. OmicSHIELD is introduced as an open-source tool aimed at overcoming these challenges by enabling privacy-protected federated analysis of sensitive omic data. In order to ensure this, multiple security mechanisms have been included in the software. This innovative tool is capable of managing a wide range of omic data analyses specifically tailored to biomedical research. These include genome and epigenome wide association studies and differential gene expression analyses. OmicSHIELD is designed to support both meta- and mega-analysis, so that it offers a wide range of capabilities for different analysis designs. We present a series of use cases illustrating some examples of how the software addresses real-world analyses of omic data.

## Author summary

OmicSHIELD revolutionizes the way researchers can engage with federated omics data, providing a secure framework for conducting different omic data analyses. This innovative platform allows data to stay in their original repositories, thus eliminating data transfer—a crucial feature in an era where data privacy regulations are becoming increasingly stringent. By leveraging advanced techniques like differential privacy, OmicSHIELD aims to mitigate disclosure risks associated with analysis of omics data, while still enabling accurate collaborative research. The platform is highly flexible, supporting processing and analysis of multiple omic data formats. This makes it a useful tool for researchers looking

are available at https://isglobal-brge.github.io/OmicSHIELD/. All data for the first use case is available on EGA under the dataset EGAD00001003338. All data for the second use case is available on FigShare at https://figshare.com/articles/dataset/HELIX_simulated_transcriptome_data/14060849. The source code of the client R package (dsOmicsClient) is available on GitHub at https://github.com/isglobal-brge/dsOmicsClient, the persistent identifier that fixes the code presented on this manuscript is the git tag '1.0.18'. The source code of the server R package (dsOmics) is available on GitHub at https://github.com/isglobal-brge/dsOmics, the persistent identifier that fixes the code presented on this manuscript is the git tag '1.0.18'.

**Funding:** This research has received funding from the Spanish Ministry of Education, Innovation and Universities, the National Agency for Research and the Fund for Regional Development (PID2021-122855OB-I00). We also acknowledge support from the grant CEX2023-0001290-S funded by MCIN/AEI/ 10.13039/501100011033, and support from the Generalitat de Catalunya through the CERCA Program and the Consolidated Group on HEALTH ANALYTICS (2021 SGR 01563). This project has also been partially funded from the "Complementary Plan for Biotechnology Applied to Health," coordinated by the Institut de Bioenginyeria de Catalunya (IBEC) within the framework of the Recovery, Transformation, and Resilience Plan (C17.I1) - Funded by the European Union - NextGenerationEU. We also thank CINECA project (EC H2020 grant 825775) for making synthetic GWAS data available through EGA repository. JU is supported by Catalan program PERIS (Ref.: SLT017/20/000119), granted by Departament de Salut de la Generalitat de Catalunya (Spain). We also thank EUCAN-Connect project (A federated FAIR platform enabling large-scale analysis of high-value cohort data connecting Europe and Canada in personalized health) funded by the European Commission H2020 Flagship Collaboration with Canada (No 824989), the ATHLETE project (Advancing Tools for Human Early Lifecourse Exposome Research and Translation) funded by the European Commission Horizon 2020 research and innovation programme (No 874583) and UKRI Innovation Fellowship with Health Data Research UK [MR/S003959/1].

**Competing interests:** The authors have declared that no competing interests exist.

to perform complex analyses across multiple datasets. OmicSHIELD includes active disclosure control checks and the ability to compute a wide range of analytical methods useful to obtain insights from omic data. By prioritizing both analytical power and data privacy, OmicSHIELD addresses the most pressing challenges in omics research today, making it easier for scientists to unlock new insights while maintaining high ethical standards.

## Introduction

Contemporary data analytics in health and biological sciences include a central focus on the analysis and interpretation of high volume 'omics data' (genomic, epigenomic, or metabolomic data). An important requirement for fully exploiting the potential of such data is to make large amounts of clinical, epidemiological and omic information accessible and interoperable to researchers. This can be achieved through data sharing. Historically, data sharing has been based on central warehousing: this requires data generators to physically transfer data or summary statistics to make them accessible to data analysts. This approach has been adopted, for instance, by most consortia devoted to the analysis of genomic data. Under this setting, each data provider runs, for example, their own genome-wide association studies (GWAS) independently, and shared summary statistics are then meta-analysed [1]. Alternatively, a recent and increasingly used analytic trend known as federated analysis (FA) permits analysis of multiple decentralized datasets, without sharing or accessing individual-level information (i.e., migrating the analysis to the data) [2]. Motivated by the delicate nature of genetic and health data and the ethical and legal issues behind sharing this kind of information, the potential benefits of FA are being increasingly widely optimized [3,4], [5].

Meta-analysis combines summary statistics from multiple studies to derive a unified conclusion. In contrast, mega-analysis (i.e. pooled analysis) in our context, uses the individual-level data from multiple studies to derive combined conclusions without the requirement of physically pooling the data. Instead, we employ specific statistical methodologies that reproduce the same results as if the data would be pooled together and analysed jointly.

Meta-analysis is widely adopted for the combination of GWAS [6] and is also being used in differential gene expression (DGE) and epigenome wide association studies (EWAS) by combining results from different populations [7]. Being capable of performing both meta- and mega-analysis in a federated framework would be a cutting-edge advance for the biomedical field, allowing, among other possibilities, to choose the best and most convenient approach to be applied depending on data characteristics and designs. Most currently available FA systems for omics data are only intended to perform pooled analyses, arguing that this approach substantially increases statistical power [8]. However, pooled analysis in a multi-cohort setting is not recommended when data are heterogeneous among cohorts or when data are not properly harmonized [9] (e.g. gene expression normalized using different methods, or GWAS data maintained on different platforms) as substantive heterogeneity in the nature of the data can lead to biased results. This includes "confounding by study" which can be very severe when an outcome and an explanatory covariate vary in tandem (or in a reciprocal manner) across study populations [10].

Omic FA has a key constraint since it includes sensitive information. It requires ensuring appropriate levels of security and privacy and the judicious application of the stringent regulations implicit to contemporary governance frameworks such as the General Data Protection Regulation (GDPR) in Europe. To address this important issue, different privacy-protecting

techniques such as federated learning (FL) in combination with differential privacy (DP) [11], homomorphic encryption (HE) [12], and secure multi-party computation (SMPC) [13] have been developed, some or all of which may be adopted [14]. Algorithms developed by researchers could potentially be used, alongside genotype-phenotype associations from genetic association studies, by an attacker to predict genotypes and phenotypes of target individuals based on genome information shared by individuals or their relatives [15], [16], [17]. A secure FA platform should thus have solutions to minimize the risk of such potential attacks.

The burden of data sharing on multi-center studies that deal with omic data has enormously increased in the last few years. These include large projects such as ORCHESTRA, MIRACUM, LifeCycle, HELIX and ATHLETE [18], [19], [20], [21], [22], [23] among many others. Having software solutions to FA in omic studies using privacy-protected techniques is therefore an urgent need. In the omic setting, infrastructures for federated networks, FA of GWAS (FAHME [8] and sPLINK [24]) and transcriptomic data (Flimma [25]) have been proposed. These existing tools have important limitations including that they may require their own data infrastructures and different programming languages–some of which are not open-source, hence making the implementation of new features difficult. Another important limitation of existing solutions is that downstream analyses (e.g. data visualization, post-omic data analyses) are poorly integrated into analytical pipelines.

In order to overcome existing limitations, we have developed OmicSHIELD which covers many types of omics data including genomics, epigenomics and transcriptomics. OmicSHIELD is designed for analysis of horizontally partitioned data, with no current support for vertically partitioned data. It is developed for DataSHIELD [2], a platform that supports secure federated analysis of sensitive data that never leaves the data owner's domain. This framework has been used to develop FA in different settings such as machine learning [26] and survival analysis [27]. It allows both mega- and meta-analyses. Two distinct data warehouses existfor the FA DataSHIELD infrastructure, Opal and Armadillo that are elsewhere described [28]. A detailed bookdown can also be accessed to see examples of how Opal can be used for data management (https://isglobal-brge.github.io/resource_bookdown/opal.html). A key feature of OmicSHIELD is that it is open-source, written in R and licensed under GPL, thus facilitating downstream analyses within a single pipeline by interacting with other programming languages (e.g. Python) and with other R or Bioconductor packages. OmicSHIELD incorporates both statisticaldisclosure controls and differential privacy approaches to assure privacy-preserving data analyses. Therefore, our approach has the potential to fulfill the stringent requirements made by data controllers (e.g. hospitals) which often hinder multicentric medical studies, since differentially private learning has been positioned as one of the preferred methods for GDPR-compliant recommender systems [28].

By enabling data to stay within its original repository and minimizing data movement, OmicSHIELD adheres to GDPR principles of data minimization and purpose limitation. Furthermore, OmicSHIELD's user-based architecture allows for audit trails and transparency, ensuring data usage is compliant with GDPR's accountability requirements.

OmicSHIELD is designed to enhance biomedical research by facilitating a wide range of omic analyses in federated settings, ensuring data privacy and compliance. In federated data analysis of sensitive omic data, addressing data privacy threats is essential. Our adversarial model focuses on "insiders with authorized access but malicious intent" recognizing the significant risk from within, often overlooked. More precisely, this falls into the honest-but-curious (HBC) adversary, which can be described as "participant who will attempt to learn all possible information from legitimately received messages" [29]. The main threat being controlled is the ability of the adversary to re-identify an individual within a dataset. These insiders, including data custodians, researchers, and technical staff, might exploit their access to infer individual-

level information from aggregated data, despite not having direct access to raw, unencrypted data. In the context of data governance, will have the opportunity tosuch action is a misuse of the data. This is why the first mitigation is always the implementation of strong data governance and the use of contractual obligations for the analyst.

To help users leverage these frameworks, we present an online book available at https://isglobal-brge.github.io/OmicSHIELD/. It covers installation, sources of help, and complete workflows illustrating examples of omic data analyses using freely available datasets. It also includes material describing different use cases corresponding to real world data applications from different existing projects. On it, readers will find further use cases than the ones described on this manuscript. Therefore, results presented in this manuscript can be fully reproduced and users can perform additional analyses using other covariates or conditions.

## Results

In this section, we initiate the discussion by presenting a broad outline of the functionalities and solutions offered by OmicSHIELD, underscoring its merits and its potential in addressing the challenges delineated in this manuscript, a summary of the key aspects is available in Table 1. Subsequently, we exemplify the practical utility of OmicSHIELD through two real-world examples of omic data analyses employing cutting-edge methodologies. In the first example, we elucidate the procedure for conducting analysis on genomic data, whereas the second example delineates the analysis of transcriptomic and epigenomic data.

### Security

There are different disclosure control methods assembled in OmicSHIELD. These can be summarized as: 1) Disclosure traps: the statistical disclosure control method of cell suppression is used to define the minimum cell sizes for functions that perform aggregation or subsettings. 2) Control of output: filtering information that the client receives, e.g. not sending the individual identifiers of a study. The behavior of these techniques can be configured by each study's data managers thus allowing individual studies to adopt a set of disclosure controls that comply with their own, local required regulations. 3) Differential privacy methods are also implemented to enable an additional layer of protection to results that are returned by study servers to the central analysis node. Differential privacy has been defined as the inability of an attacker to distinguish whether a single individual is present in a dataset [30]. Adding stochastic noise to function outputs is a way of achieving this: different types of noise can be used [31], with fine-tuned Laplace noise being the chosen method for our implementation (see Methods section). This approach has been adopted as a countermeasure to inference attacks using complex queries [32], [33], [34]: as such attacks can make use of GWAS results and allele frequencies,

Table 1. Key aspects of OmicSHIELD.

| Key issue for omic FA | OmicSHIELD solution |
|---|---|
| Different types of federated analyses | Pooled analysis and meta-analysis |
| Non-disclosive analyses | Those by DataSHIELD |
| Privacy-protected / attacks | Differential privacy, filters, audit activity |
| Different omic data | Functions for genomics, transcriptomics and epigenomics with easy extension to metagenomics |
| Open-source | GPL3 and MIT license |
| Interaction with other tools | Post omic analyses with R/Bioconductor, other FA for clinical or epidemiological analyses with DataSHIELD |

the implemented differential-privacy mechanisms are intended to cause additional difficulty for such attacks. 4) There are other strategies that can be performed using minor allele frequencies (MAF) attacks [35]. To prevent these attacks, we offer an extra layer of protection by having a filter that blocks the output for SNPs with a MAF lower than a pre-specified threshold. This threshold is also configurable by data managers. Furthermore, the OmicSHIELD architecture is user-based, meaning the activity of authorized researchers can be audited to check whether inference attacks have been performed. OmicSHIELD has all filters active by default with sensible values (i.e. $\varepsilon = 3$, resample = 3, MAF = 0.05), nevertheless it is up to the data managers to modify these values according to their needs or privacy budgets.

## Omic analytic capabilities

OmicSHIELD contains functionalities to perform three types of omic data analysis: GWAS, DGE and EWAS. Table 2 outlines the main functions available in OmicSHIELD. Fig 1 demonstrates how omic association analyses can be performed using client-side functions (i.e. using the dsOmicsClient package). Data (omic and phenotypes/covariates) are stored in their native formats at different sites (for example, remote servers accessible via https or ssh, Amazon S3, locally,. . .). Analysis follows the DataSHIELD client-server architecture, implemented through a pair of libraries with dsOmics implemented server-side and dsOmicsClient client-side. Most association analyses involving omic data are based on fitting different generalized linear models (GLMs) for each feature (e.g., SNP, CpG, gene, transcript, . . .) and this forms the basis of the methods we have implemented, which includes two different types of analysis: pooled and meta-analyses.

The "pooled approach" (Fig 2A) is recommended when the researcher wants to analyse omic data from different sources and obtain results as if the data were physically pooled in a single database. The algorithm for pooled regression analysis can be very time consuming for omic data, therefore we implemented a fast algorithm for massive generalized linear models (see Methods). This approach is not recommended for data that is not fully harmonized–it should not be performed when gene expressions are normalized using different methods, or when GWAS is provinent of different data collection technologies, as substantive heterogeneity in the nature of the data can lead to biased results [36]. The "meta-analysis approach" (Fig 2B) overcomes limitations raised when performing pooled analyses. Computation issues are addressed by using scalable and fast methods to perform data analyses at the whole-genome level at each server. Best practice methods to perform meta GWAS should be adopted [37]. We have implemented these methodologies in OmicSHIELD. Adopting a meta-analytic approach is also recommended when there is large heterogeneity in the trait of interest between cohorts.

## Genomics, polygenic risk scores and principal components analyses

Genomic data are analysed using the GWASTools and GENESIS Bioconductor packages that allow quality control (QC) and GWAS to be performed [38]. As we can also use computational resources [39], we have implemented methods to perform meta-analyses using PLINK and SNPTEST which are standardly used to perform GWAS using genotyped and imputed SNPs, respectively.

We use the PGS Catalog to calculate polygenic risk scores (PRS) from curated literature [40]. As this information can be disclosive, OmicSHIELD calculates PRS for each individual at each server without any interaction between the cohorts. The PRS are stored on the servers and are considered thereafter as any other covariable. That is, PRS can be used as part of an association model.

**Table 2. Main analysis functions of OmicSHIELD.** For the complete list of functions and the complete details refer to the available online guide (https://www.isglobal-brge.github.io/dsOmicsClient).

|  | Function | Description |
|---|---|---|
| **Genomics** | eafPlot | Plot the reported estimated allele frequencies against a reference set (1000G) |
|  | lambdaNPlot | Plot to reveal issues with population stratification |
|  | pzPlot | Plot to reveal issues with beta estimates |
|  | seNPlot | Plot to reveal issues with trait transformations |
|  | ds.fastGWAS | Performs a pooled fast GWAS using the algorithm described in the "Methods" section |
|  | ds.metaGWAS | Performs a meta-analysis GWAS using the GENESIS Bioconductor library |
|  | ds.alleleFrequency | Calculates the allele frequencies. Can be used pooled or as a meta-analysis |
|  | ds.exactHWE | Calculates the exact HWE test using Fisher's method. There is the option of only using the controls to calculate this test |
|  | ds.PCA | Performs a pooled PCA using only the SNPs that have been linked to differentiate ethnic groups |
|  | ds.PRS | Calculates the polygenic risk scores of the individuals sourcing the risk SNPs and weights on the PGSCatalog |
|  | ds.PLINK | Creates a remote connection to a machine with PLINK to remotely run analysis commands using traditional PLINK syntax |
|  | ds.snptest | Creates a remote connection to a machine with SNPTEST to remotely run analysis commands using traditional SNPTEST syntax |
|  | manhattan | Plots a Manhattan plot using the results from ds.fastGWAS and ds.metaGWAS |
|  | LocusZoom | Plots a LocusZoom plot using the results from ds.fastGWAS and ds.metaGWAS. It can retrieve the genes present on the region of interest using BioMaRT and TxDb.Hsapiens.UCSC.hgXX.knownGene (XX can be 37 or 38) |
|  | plotPCA | Plots the results of ds.PCA. The plot can be color coded using categorical variables of the genomic data |
|  | ds.genoDimensions | Get dimensions of the genotype: number of SNPs, scans and chromosomes |
|  | ds.getChromosomeNames | Get the names of the chromosomes |
|  | ds.getSNPSbyGen | Subset genotype by a gene |
|  | ds.getSNPs | Get available SNP names |
|  | ds.glmSNP | Logistic regression for single SNP |
|  | ds.table_gds | Get contingency table of phenotypes |
| **Other Omics** | ds.addPhenoData2eSet | Auxiliary function to add phenotype data to the ExpressionSets that contain the omic data |
|  | ds.createRSE | Create a RangedSummarizedExperiment from a counts table and a phenotypes table |
|  | ds.featureNames | Get the names of the features |
|  | ds.featureData | Get information about the features |
|  | ds.fvarLabels | Get the names of the phenotypes |
|  | ds.nFeatures | Get number of features |
|  | ds.nSamples | Get number of samples |
|  | ds.RNAseqPreproc | Determine which genes have sufficiently large counts to be retained in a statistical analysis |
|  | ds.limma | Fits a limma + voom model |
|  | ds.edgeR | Fits an edgeR model |
|  | ds.DESeq2 | Fits a DESeq2 model |
|  | ds.removeOutliers | Remove potential outliers using winzorization |
|  | ds.subsetExpressionSet | Subset by categorical phenotype |
| **Meta analysis** | metaBetaValues | Performs meta-analysis of beta values |
|  | metaPvalues | Performs meta-analysis of p-values using the sum of logs method (Fisher's method) |
|  | qqplot | P-values qqplot |

Computing principal components analyses (PCA) is the standard methodology to address populations in GWAS, but computing federated PCA is missed in other federated GWAS solutions (e.g. FAHME [8] or sPLINK [24]). Existing approaches use principal components (PCs) estimated at each cohort and these covariates are then used for the adjustment of association models. However, PCs should be computed using the entire population to capture genetic

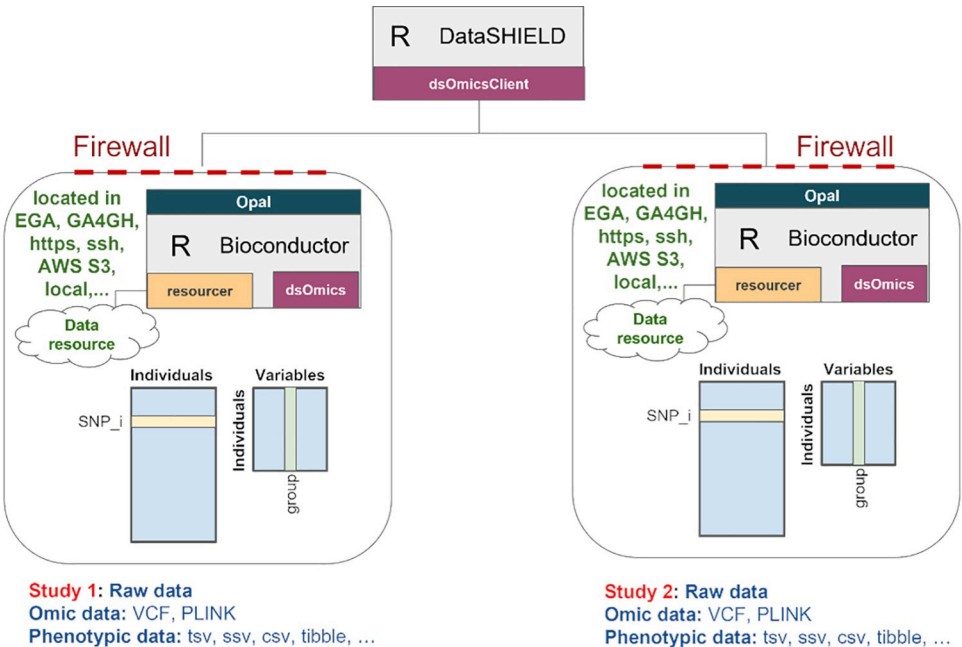

**Fig 1. Scheme of DataSHIELD implementation of omic-related packages.** The *dsOmics* package contains functions to perform non-disclosive data analyses of omic data that are managed within Opal using the *resourcer* package. Omic data normally have two pieces of information, one corresponding to features (CpGs, SNPs, genes, . . .) and another for phenotypic data (grouping variable, outcome, covariates, . . .) that can be stored in different resources (e.g. PLINK and table in genomics) or in a specific resource designed for that purpose in R/Bioconductor (e.g. *ExpressionSet* or *RangedSummarizedExperiment*). This package should be installed in the Opal server along with their dependencies. The package *dsOmicsClient* must be available in the client side and contains functions that allow the interaction between the analysis computer and the servers.

differences among individuals [41]. OmicSHIELD can circumvent this issue by adopting a pooled approach, thus providing a better solution than any other available elsewhere.

## Differential gene expression analysis and EWAS

The DGE and EWAS meta-analyses provided by OmicSHIELD make use of the widely used Bioconductor ExpressionSet, RangedSummarizedExperiment or GenomicRatioSet data formats to deal with omic and phenotypic (e.g. covariates) information. The main analysis method implemented corresponds to RNA-seq data which are analysed using limma+voom [42]. We have also implemented functions for using other methods such as DESeq2 and edgeR [43] as well as methods to analyse microarray data using limma.

## Post-omic analyses and visualization

The pooled approach returns the effect sizes, standard errors, and some specific feature annotations. On the other hand, the meta-analysis approach provides study-specific estimates and standard errors from analyses performed on each server. These results can then be brought together using various meta-analytic techniques. For example, GWAS utilizes effect sizes and standard errors, while DGE and EWAS employ p-values [7]. Both methods are incorporated in OmicSHIELD.

Various visualizations can be created after the completion of an analyses. For example, Manhattan and qq-plots can be generated for all types of analysis, while a locus zoom plot can

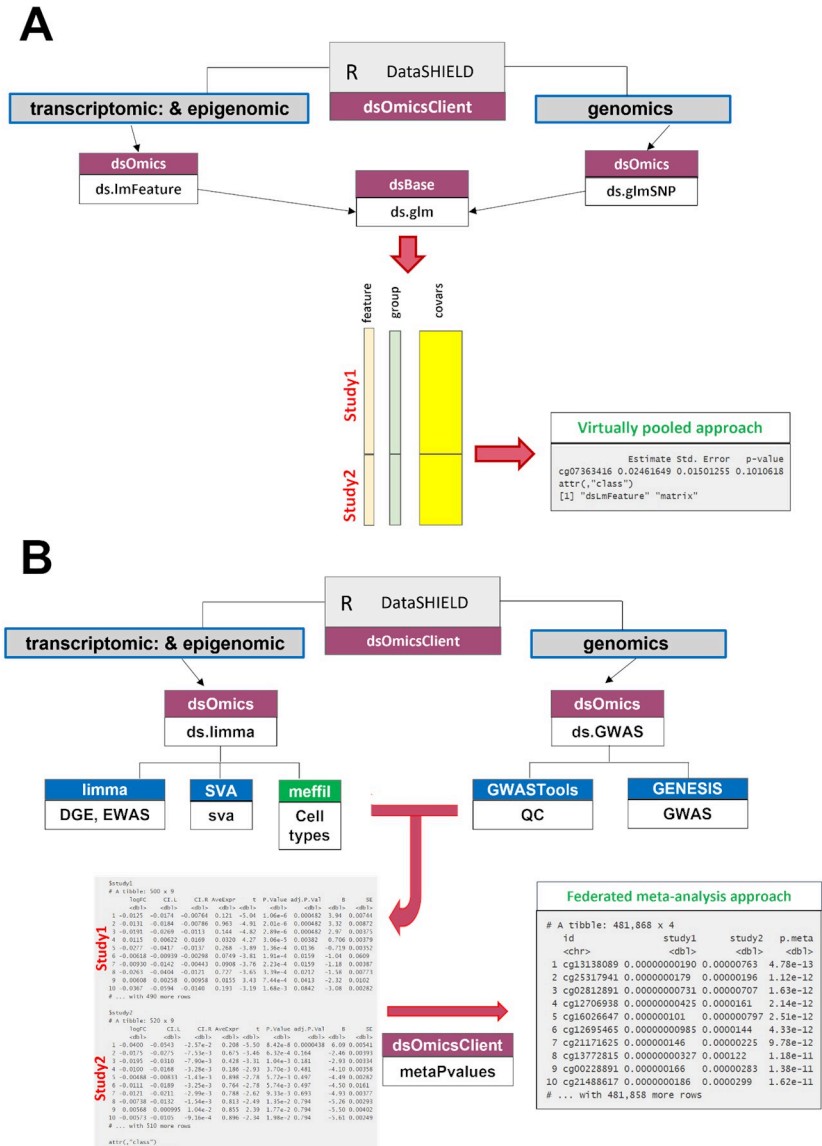

**Fig 2. Types of omics data analyses implemented in *dsOmicsClient*.** We have implemented two different types of analyses: the virtually pooled and the federated meta-analysis. **Panel A** shows the "virtually pooled approach" that is recommended when the user wants to analyse omic data from different sources (e.g. studies) and obtain results as if the data were in a single data warehouse. **Panel B** depicts the "federated meta-analysis approach" that overcomes the limitations raised when performing pooled analyses: computing time and data harmonization. The computation issue is addressed by using scalable and fast methods to perform data analysis at whole-genome level at each server. The data harmonization issue is addressed by combining results using p-values that are independent of how omic data in features have been recorded.

be made for GWAS. Visualization of data distributions for any feature grouped by a specific variable (e.g. using boxplots) can be performed for DGE and EWAS.

## Use case 1: Multi-centric GWAS of CINECA data

To evaluate our new approach to GWAS analysis, we used a public dataset of synthetic genotype data from CINECA (see Data availability statement). This dataset has been split into three

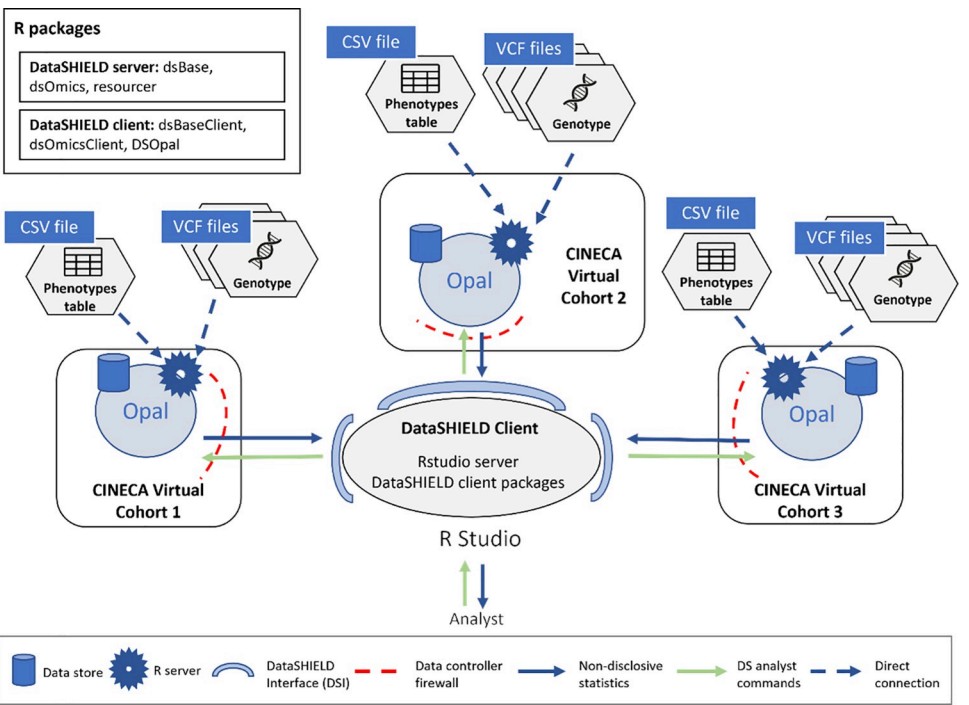

**Fig 3. Configuration for multi-centric GWAS of CINECA data using OmicSHIELD.** To achieve this configuration, the data from CINECA has been partitioned into three different virtual cohorts, containing 817, 1073 and 614 individuals respectively. Each virtual cohort contains the genotype information of the individuals. The amount of variants present for each individual is 865 thousand.

different subsets and uploaded to three virtual servers to act as individual study centers. The virtual configuration is illustrated in Fig 3.

We compare the results obtained using OmicSHIELD with those obtained by pooling the three datasets into a single dataset and being analysed following a traditional local pipeline. We are interested in assessing associations between SNPs and diabetes, information that is obtained from a variable called 'diabetes_diagnosed_doctor'. We adjust for other covariates including sex, age and high-density lipoprotein (HDL) cholesterol. The effect sizes (i.e. beta values) of the top 20 SNPs obtained with OmicSHIELD are compared with the effects obtained from the traditional pipeline. This comparison yielded a mean square error of $5.3 \times 10^{-4}$ and a bias of $-2.6 \times 10^{-3}$ which is almost negligible in practical terms (bias in the risk of a given SNP is to the order of 1 in 10000). The Manhattan plot in Fig 4 shows that the top hits among the p-values and the general trend of significance levels are accurately replicated using Omic-SHIELD. We can see that the noise added by the differential privacy method ($\varepsilon$-privacy: 3) allows trends to be replicated while ensuring that the top hits remain significant. Please refer to the data availability statement section to reproduce the results presented.

After demonstrating the efficacy of OmicSHIELD in replicating the significance of p-values and identifying top hits, we further investigate the influence of differential privacy parameters on the usability and interpretability of our results. Differential privacy employs a privacy parameter, epsilon ($\varepsilon$), to balance data utility against privacy. A smaller $\varepsilon$ value indicates higher privacy but potentially less accurate results, while a larger $\varepsilon$ allows for greater accuracy at the cost of privacy. To quantitatively assess this trade-off, we conducted additional analyses varying $\varepsilon$ values at 0.01, 0.05, 0.07, 0.09, 0.1 and 1. These experiments aim to portray how changes in $\varepsilon$ affect the overall utility of our federated analysis results, particularly in terms of

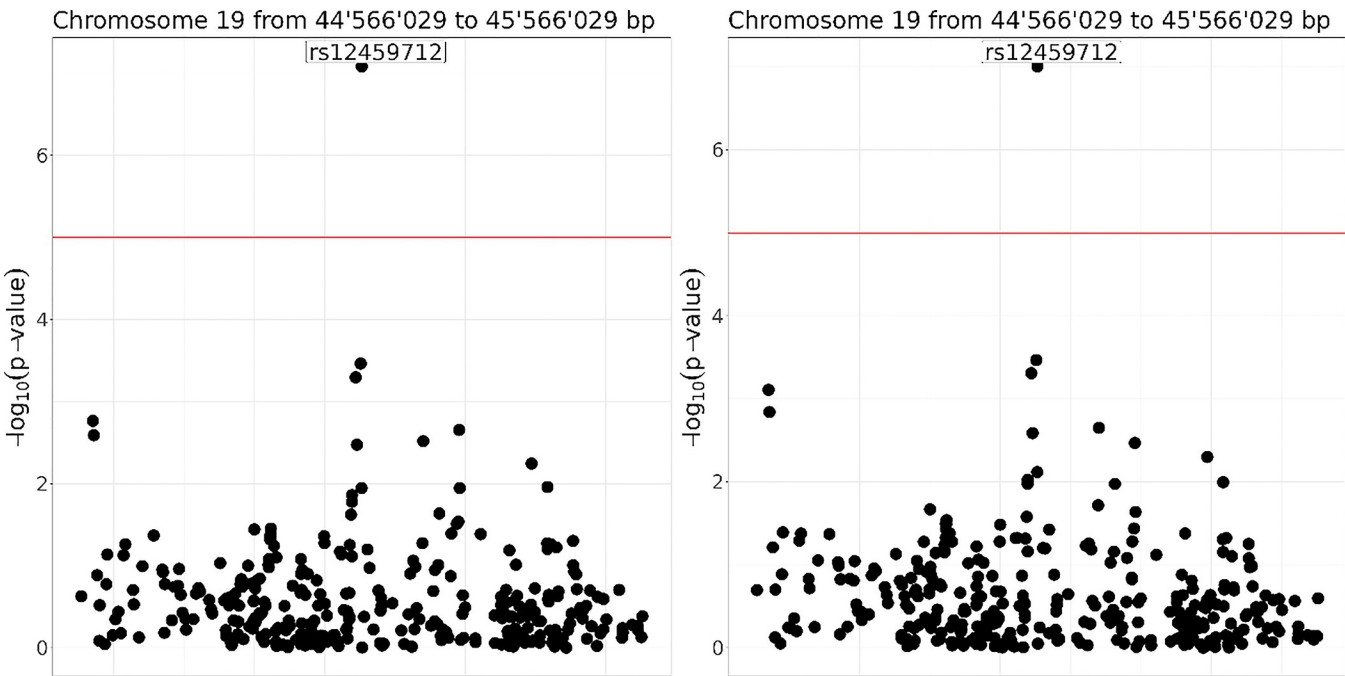

**Fig 4. Locus zoom plots of the top hit for the original data (left) and pooled fast GWAS (right).** The red line corresponds to a threshold of significance of -log10(P) $< 5 \times 10^{-5}$. Trends and top hits are reproduced on the OmicSHIELD analysis, the major differences being found on the SNPs clearly not relevant.

statistical power and the ability to detect significant associations. The outcomes of these analyses are shown in Fig 5.

We observed a notable threshold at ε = 0.09, which emerges as a critical point delineating the trade-off between maintaining privacy and achieving reliable statistical results. Below this ε value, our analysis revealed an increased incidence of false significant hits. Conversely, when ε values are set above this threshold, the stability and reliability of the results significantly improve. This stability indicates that ε = 0.09 represents an optimal balance according to the authors findings. These findings underscore the importance of carefully selecting differential privacy parameters in federated omic data analyses to optimize both privacy protection and data utility.

## Use case 2: DGE and EWAS analysis of HELIX data

Here we illustrate how to perform DGE and EWAS of HELIX data (see Data availability statement). The data infrastructure for this project is shown in Fig 6. Transcriptome and epigenome data are stored as ExpressionSet, which is a standard Bioconductor infrastructure to deal with these types of omic data [44]. Note that this Bioconductor object contain phenotypic data (i.e. metadata) encapsulated jointly with the omic data. In both examples, we are interested in comparing gene expression and methylation between males and females focusing only on the autosomes.

For the DGE analysis, we analyse microarray data derived using the "Human Transcriptome Array 2." of Affymetrix. Among the different analyses, we show how to apply meta-analysis with the functions ds.limma() and metaPvalues(). For each microarray probe, this analysis implements multiple generalized linear models separately (one per study) and combines the results using study-specific derived p-values. As a result, we identified a list of 325 probes (mapping 287 genes) differentially expressed between boys and girls of the HELIX project, and

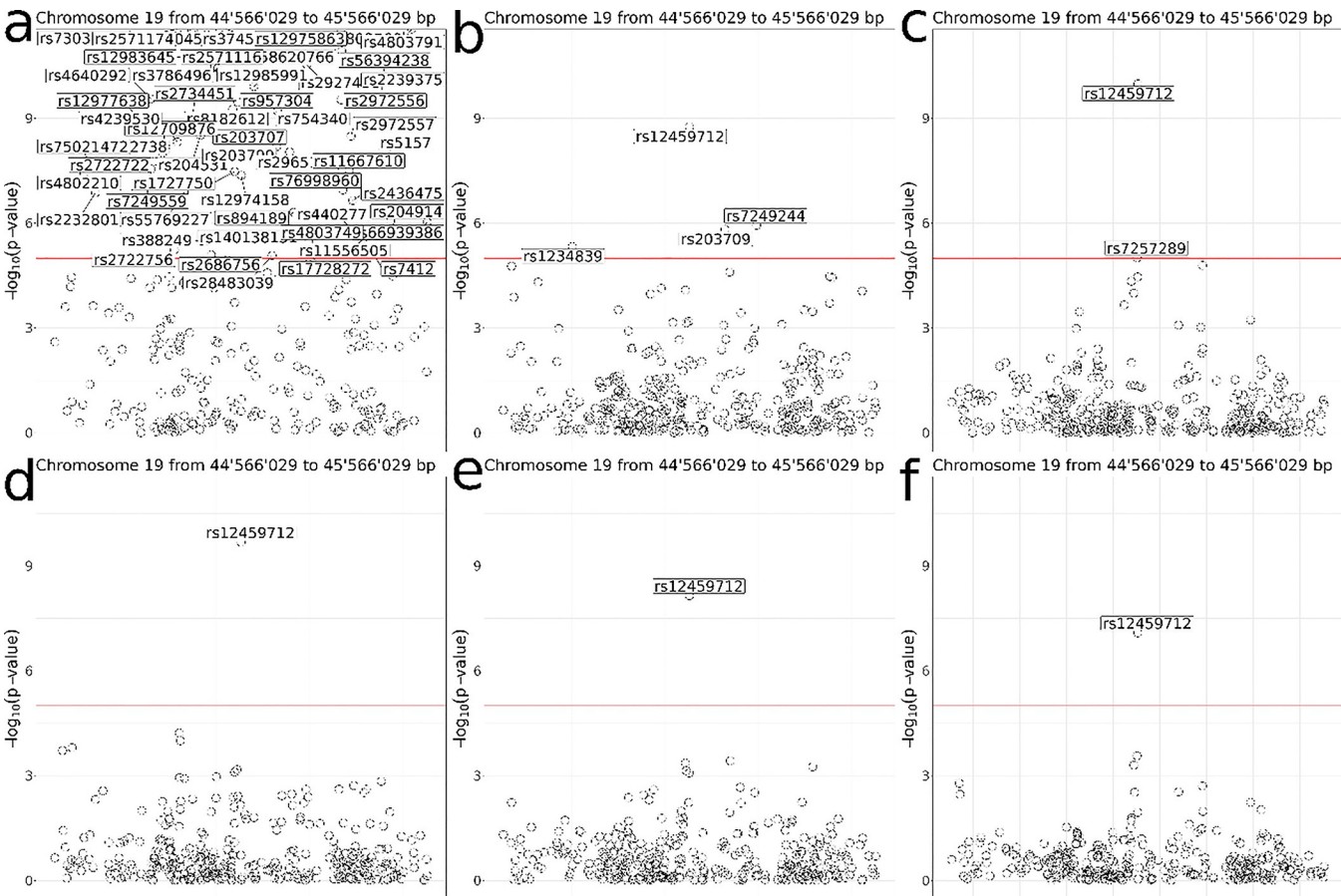

**Fig 5. Impact of Differential Privacy Parameter (Epsilon) on the Usability of Federated Analysis Results.** This figure presents the analysis of the effect of varying epsilon (ε) values on the detection of significant genetic associations in a federated omic data setting. Epsilon values tested include 0.01 (a), 0.05 (b), 0.07 (c), 0.09 (d), 0.1 (e) and 1 (f).

which further passed multiple-testing correction filters (P-value threshold = 1.74e-06). As a proof of the ability of OmicSHIELD to be integrated with other R functionalities and Bioconductor packages, we continued the pipeline presenting results of a functional enrichment analysis (FEA). This analysis shows that significant differentially expressed genes participate in processes with evident and previously-described sexual dimorphism such as in the case of "Longevity regulating pathways" (KEGG Pathway:Ia04211).

For the EWAS data we illustrate how to perform DNA methylation differential analysis using microarray data obtained with the "Infinium HumanMethylation450" platform of Illumina. We illustrate how to perform an epigenome-wide meta-analysis with and without adjusting for surrogate variables. We also adjusted our models for confounders including age and ethnicity. Consequently, from the initial list of almost 300k CpGs, we identify a total of 10,417 differential methylated probes between boys and girls from which only three passed the strict Bonferroni multiple-testing correction. Interestingly, two of these three probes, cg12052203 and cg25650246 (mapping the B3GNT1 and RFTN1 respectively), have been previously associated with sex methylation differences (http://www.ewascatalog.org). The FEA showed that significant CpGs map genes participating in processes with strong sex differences such as in the case of bone formation ("Endocrine and other factor-regulated calcium reabsorption", KEGG PathwI hsa04961). Please refer to the data availability statement section to

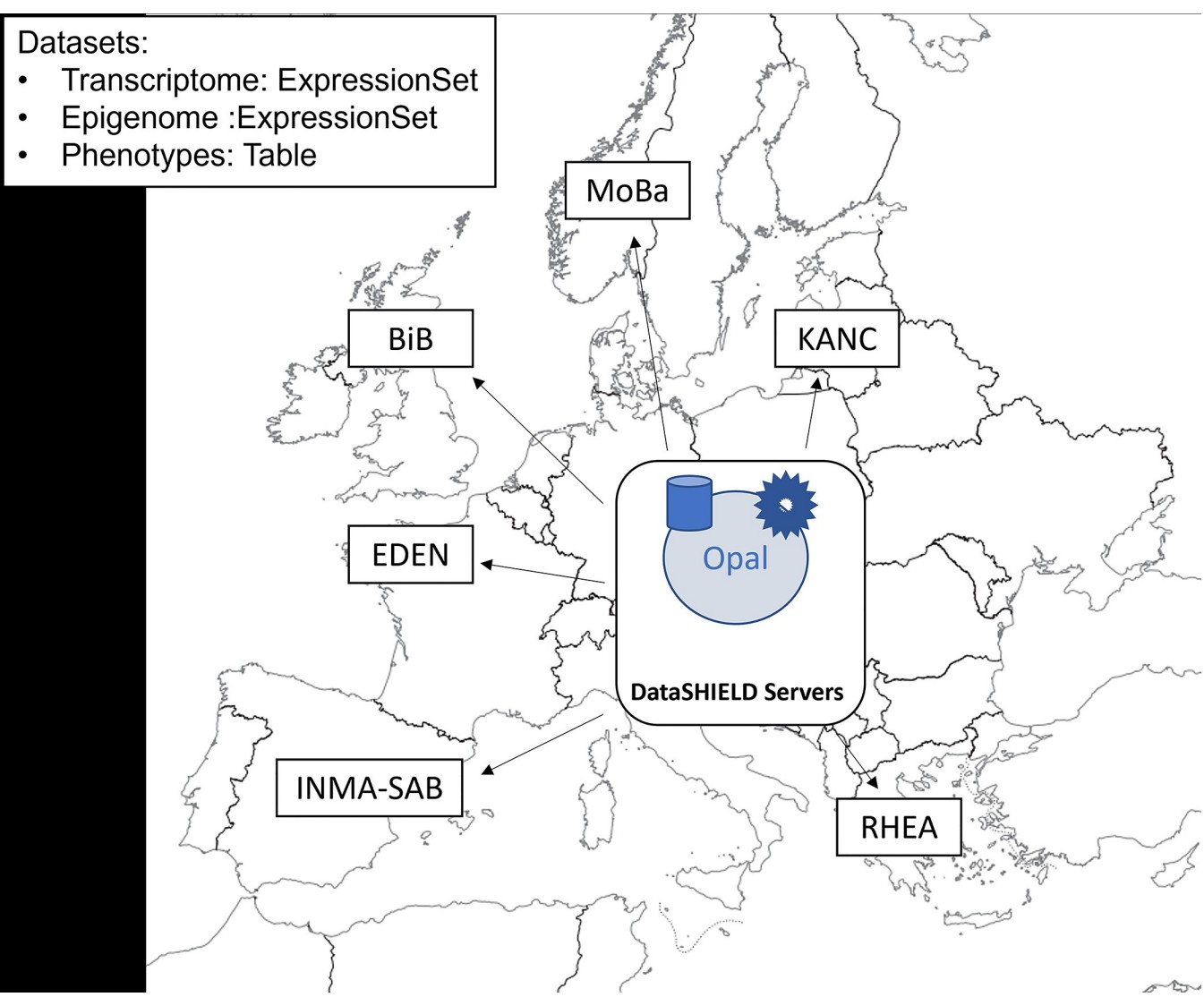

**Fig 6. Opal Data infrastructure of HELIX project.** The raster data used to generate the map comes from https://www.naturalearthdata.com/.

reproduce the results presented. For the sake of reproducibility, we included all the data on the same server, then different connections to it are performed, this replicates the same situation as having different servers scattered across different countries. The differential privacy settings used on this use case is $\varepsilon = 3$.

## Validation and Application of OmicSHIELD using Real-world Methylation Data

The OmicSHIELD infrastructure has been presented as a robust, privacy-protected and open-source analytical platform for federated meta- and mega-omic data analysis in multi-centre studies. To demonstrate the utility and validity of OmicSHIELD, a researcher conducted a real-world case study using methylation data from two independent cohorts of the ATHLETE project, INMA (INfancia y Medio Ambiente) and EDEN (Etude des Déterminants pré et post-natals précoces du développement et de la santé de l'Enfant).

The study aimed to validate OmicSHIELD using INMA methylation data by comparing the EWAS results from both DataSHIELD and local analyses, without implementing differential privacy. This comparison was essential to determine if the results obtained through the OmicSHIELD platform were consistent with those derived from conventional local analysis methods. Upon validation of the consistency and accuracy of the results, the researcher proceeded to use DataSHIELD for the analysis of the EDEN cohort.

The analysis of the INMA methylation data using OmicSHIELD yielded aggregated EWAS results that were in accordance with the results obtained through local analysis, thus demonstrating the reliability and validity of the platform. This validation provided confidence in utilizing OmicSHIELD for further data analysis while maintaining data privacy and adhering to ethical guidelines. Given that this validation was performed under no differential privacy mechanisms, the regression estimates for each model were identical for both approaches (i.e. with and without using OmicSHIELD).

Following the successful validation, the researcher proceeded to analyse the EDEN cohort data exclusively through DataSHIELD, incorporating the results into their publication, which is still being elaborated. The application of OmicSHIELD in this real-world case scenario showcases its potential as a valuable tool for researchers in the field of bioinformatics, allowing for privacy-protected federated data analysis across multiple centres without compromising the integrity of the results.

In conclusion, the validation and application of OmicSHIELD using real-world methylation data from INMA and EDEN cohorts provide strong evidence for its effectiveness and reliability as an open-source analytic platform for privacy-protected data analysis in multi-centre studies. The successful implementation of OmicSHIELD in this real study highlights its potential to significantly advance the field of bioinformatics and facilitate novel discoveries in the omics sciences while mitigating data disclosure risks.

## Discussion

We have presented a new software to perform omic analyses using multi-center studies (i.e. federated data) with active disclosure protection during analysis and for the outputs. Such a tool, based on the paradigm of DataSHIELD, provides a great opportunity for researchers to enhance multi-center collaboration by establishing a trustworthy platform that brings the analysis to the data, hence avoiding onerous data-sharing procedures. The solution we present is fully open-source, enabling researchers not only to contribute their own developments, but to personally assess and control the disclosure-prevention and differential-privacy mechanisms. This guarantees the data contributors (e.g. research participants) and custodians (e.g. data controllers) complete transparency on how data is utilized, something that is central to the philosophy of the GDPR but is not achievable with commercially licensed softwares.

OmicSHIELD has implemented state-of-the-art methods in GWAS, transcriptomic and epigenomic data analysis including new methodologies. For example, for GWAS, we have implemented quality control of individual studies before performing pooled- or meta-analysis [37], and non-disclosive pooled principal component analysis (PCA) to adjust for population stratification in pooled GWAS (something not addressed in FAHME [8] or sPLINK [24]). For transcriptomic and epigenomic studies, we have implemented outlier removal and surrogate variable analysis, and differential expression analyses using not only limma and voom but also other existing approaches, including edgeR and DESeq, that could be also used for metagenomics data analyses [43].

OmicSHIELD provides the results one would expect when having all the data combined on a single machine, however, the data are neither shared between study servers nor stored on an

intermediate server. Simply stated, data never leaves the study centers where they are hosted and thus remain fully controlled by their data owners. Pooled analysis can be beneficial to improve statistical power. However, this approach is not useful when huge imbalances exist between different datasets. Our software solution includes approaches for researchers to select the best methods taking into consideration how data has been collected or harmonized, different types of study designs, and data heterogeneity. Both implemented methods are privacy-protected using disclosure traps (based on staistical disclosure control methods) and differential privacy. The disclosure traps and differential privacy are configurable at the data source level; the decision of whether to apply differential privacy (and it's ε-differential privacy) is dictated by the data manager of each site, and it is possible to have different sites using different configurations, thus offering flexibility to multi-center studies where there are different views and regulations on data protection and disclosure risks.

Given the proposed solution relies on differential privacy, it could be discussed that there has to be a system in place that limits the number of queries. That is indeed true, although it is not a mechanism that has to be included into OmicSHIELD, but rather a mechanism provided by the DataSHIELD architecture, given that all the different analysis packages designed for it would benefit. While this is not yet available, but there is active logging of how the users interact with the federated platform, so an active control over the authorized users is possible to take into action.

Currently, there are several International projects that have set up an infrastructure using DataSHIELD to perform FA. These include UnCoVer [19], ATHLETE [21], LifeCycle [20], and InterConnect [45], in addition to national consortia established in Germany (e.g. INTI-MIC [46] and MIRACUM [47]) and Sweden [48]. They started by analysing data addressing clinical and epidemiological scientific questions but are now also moving towards including omic data analyses. In this regard, OmicSHIELD will provide a great solution for performing FA in these large consortia and allow the examination, for instance, of the impact of the exposome on the epigenome, discovering new genetic risk factors for persistent COVID or knowing how the exposome impacts human health, among others.

The presented iteration of OmicSHIELD has the potential to reproduce many published papers as well as be the main driver of new investigative projects; nevertheless, there are many ways this software could be expanded in the future. Discussion to determine future directions to be taken will be held with researchers that use our tool for their work; in this way, we can guarantee that future versions of OmicSHIELD will contain functionalities required by real-life projects, ensuring its longevity and quality. Besides having access to new developments of OmicSHIELD, researchers that choose to use the DataSHIELD ecosystem will also benefit from the growing array of available open-source libraries (https://www.datashield.org/help/community-packages), thus enabling them to use a wide variety of tools to perform non-disclosive statistical analyses on their data.

## Methods

OmicSHIELD is based upon our recent development, the "resources" architecture which is a new DataSHIELD tool that allows: 1) the use of large data in their original repositories; 2) working with original data formats (e.g. PLINK, VCF, ExpressionSet, RangeSummarizedExperiments); 3) interactions with other programming languages (including shell commands) and software (R, Neuroconductor Bioconductor, Python); and 4) interfacing with "Apache Spark", a fast and general purpose analytical engine for big data and deep learning [39]. Along this section we describe some of the algorithms, there are two entities mentioned data processors (DPs) and the client. The DPs are the servers where the data is stored and the analysis is

performed. The client, typically represented by the researcher, sends instructions to the DPs for data analysis and receives back the results

## Differential privacy

Differential privacy can be briefly described as an algorithm that prohibits individual level information from being identified when an output is observed. This concept was introduced by Dwork [49]. Formally, this concept is referred to as $\varepsilon$-differential privacy and is mathematically expressed as:

$$\Pr[A(D_1) \in S] \leq \exp(\varepsilon) \cdot \Pr[A(D_2) \in S]$$

Where A is a randomizing algorithm that takes a dataset D as input, and $\varepsilon$ is a positive real number, called the privacy parameter, that defines the level of privacy (closer to 0 indicates more privacy). S refers to all subsets of the image of A i.e. all possible subsets of the output values of the algorithm A. The inequality refers to datasets $D_1$ and $D_2$ that differ on a single element (i.e., the data of one person). If met, it indicates that after application of the randomizing algorithm, the probability that $A(D_1)$ lies in S is less than or equal to the probability that $A(D_2)$ is in S multiplied by the constant $\exp(\varepsilon)$ that gets closer to 1.00 as $\varepsilon$ falls to 0.

In order to implement differential privacy into our software, we used the Laplace mechanism. This adds noise to the output of a function, the noise being drawn from a Laplace distribution with mean 0 and scale $\frac{\Delta f}{\varepsilon}$, where $\Delta f$ is the l1 norm defined by $\Delta f = max_{D_1, D_2} \|f(D_1) - f(D_2)\|$.

In practical terms, we perform data anonymization through addition of Laplace noise (with zero mean and variance equal to a proportion of the initial variability). We also propose the removal of the noise effect at the analysis stage (to increase the accuracy of the model estimates) through statistical techniques of measurement errors. In the case of GWAS, this method guarantees privacy at variant-level, the disclosure risk could be increased on the event that multiple variants are correlated; given the adversarial model of our software this case is out-of-scope of this research, as DataSHIELD users are required to adhere to established data governance rules for data use. Our software enforces these rules, thereby mitigating further risks.

The assessment of $\Delta f$ can be particularly complex for certain functions (e.g. limma + voom), hence a resampling method [50] is used to assess $\Delta f$. The used resampling method estimates the sensitivity to achieve random differential privacy [51]. This method introduces an additional risk parameter to the differential privacy, $\gamma$, which represents the probability that the privacy loss can exceed $\varepsilon$. Following the conditions defined in Theorem 15 of [50] it is used to compute the number of resamples (N) to guarantee differential privacy under this assumption. A lower probability of privacy loss exceeding $\varepsilon$ (low $\gamma$) will translate to a higher N. The privacy parameters can be configured by the data manager of each study while the default values are set to $\varepsilon = 3$ and $\gamma = 0.1$. This method implies that differential privacy is slightly weaker than traditional differential privacy (where sensitivity is assessed analytically) due to the probabilistic nature of its guarantee. However, by choosing a sufficiently small $\gamma$, the practical difference in privacy protection becomes negligible for most applications. This resampling method has been used on all the analysis methods that return results to the client. In order to implement the layer of differential privacy to OmicSHIELD, we add the calculated noise to the results that are returned to the client. The resampling algorithm works as follows:

*A*: Function to which differential privacy is to be applied

*D*: Complete dataset

$D_1$, $D_2$: Dataset missing a random individual

N: Number of resamples
Step 1: Derive $D_1$, $D_2$ randomly from D
Step 2: Compute $A(D_1)$
Step 3: Compute $A(D_2)$
Step 4: Compute $\Delta fN = \|A(D_1) - A(D_2)\|$
Step 5: Redo steps 1–4 N times
Step 6: $\Delta fN = \max(\Delta f_{1,2,\ldots,N})$

## Allele frequency

To compute the allele frequency using horizontally partitioned data, we use the following algorithm, where the multi-layered security approach of OmicSHIELD can be appreciated, precisely on steps 1.2 to 1.4.

$DP_i$: Data processor (Server node)
geno: Encoded genotype data
Step 1: Intermediate allele frequency
 1: Each $DP_i$ computes allele_frequency(geno) $[MAF_i, n_i]$
 2: Each $DP_i$ computes differential privacy $\Delta f$
 3: Each $DP_i$ applies diff_privacy(MAF_i) $[MAF\_i]$
 4: Each $DP_i$ checks $MAF_{filter}$ $(MAF_i)$ $[MAF_i]$
 5: Each $DP_i$ returns $MAF_i$ to the client
Step 2: Client aggregation
 1: The client receives $MAF_i$ and $n_i$
 2: The client sums $MAF_i$ and divides by the sum of $n_i$
 3: The client returns the pooled MAF results [final_results]

## Block method to calculate pooled PCA

To compute a pooled PCA using horizontally partitioned data, the block method singular value decomposition has been implemented [52] [53]. The pseudo-code of the algorithm is as follows:

$DP_i$: Data processor (Server node)
SVD: Singular value decomposition
geno: Encoded genotype data
Step 1: Intermediate SVD
 1: Each $DP_i$ computes SVD(geno) $[u_i, v_i, d_i]$
 2: Each $DP_i$ computes $u_i * d_i$ $[res_i]$
 3: Each $DP_i$ returns $res_i$ to the client
Step 2: Client aggregation
 1: The client receives $res_i$ and merges it horizontally [aggregated]
 2: The client computes SVD(aggregated) [final_results]
 3: The client returns the pooled PCA results [final_results]

The algorithm used returns the product of the left singular vectors ($u_i$) and singular values ($d_i$) to the client. This guarantees that the original data contained on each node are not reconstructible (i.e. we only share non-disclosive information among servers). Moreover, differential privacy is applied to the results that the client receives following the aforementioned methodology.

This is a general method to compute PCA. We have adapted this approach to allow PCA to be performed on genotype data. To this end, these two extra steps have been implemented:

- Genotype standardization: $\frac{X_{ij} - 2\mu_j}{\sigma_j}$, where the genotype is $X_{Ij}$ (i is the individual index, j is the SNP index), $\mu_j = 2p_j$ is the mean of the 0,1,2-coded genotype ($p_j$ being the alternate allele frequency), and $\sigma_j$ is the expected standard error under Hardy-Weinberg Equilibrium, that is $\sqrt{2p_j(1-p_j)}$.

- Perform PCA using only selected SNPs that have been linked to differentiate ethnic groups [54].

## Fast virtually pooled GWAS

The method implemented for fast GWAS is based on research by Sikorska et al[55] using an algorithm adapted to our infrastructure. This method is used to perform GWAS on a pooled setting. Our implementation is described in the following pseudo-code:

$DP_i$: Data processor (Server node)

Step 1: Fit the objective model *

 1: Iteratively each $DP_i$ extracts the coefficients of objective model

 2: Client sends to each $DP_i$ the coefficient values

 3: Each $DP_i$ computes the fitted.values and residuals [$FIT_i$, $RES_i$]

Step 2: Using the residuals and genotype information [$RES_i$, $GEN_i$]

 1: Each $DP_i$ computes $RES_i$–mean ($RES_i$): $> YC_i$

 2: Each $DP_i$ computes colSums($YC_i * GEN_i$); colSums($GEN_i$); colSums($GEN_i^2$)

and returns to the client [$B_i$, $S1_i$, $S2_i$]

 3: The client merges [$B_i$, $S1_i$, $S2_i$] into [$B$, $S1$, $S2$]

 4: The client computes the total number of individuals: $> N_{IND}$

Step 3: Using [$B$, $S1$, $S2$, $N_{IND}$, $YC_i$] compute betas and pvalues

 1: The client computes $S2 - (S1^2)/N_{IND}$: $> DEN1$

 2: The client computes $B/DEN1$: $> \beta$ to obtain the beta values

 3: Each $DP_i$ computes colSums($YC_i^2$) and returns to client: $> YC2_i$

 4: The client merges $YC2_i$: $> YC2$

 5: The client computes $(YC2 - \beta^2 * DEN1)/ - (N_{IND} - K - 2) :> \sigma$

 6: The client computes sqrt($\sigma * (1 / DEN1)$): $> ERR$

 7: The client computes 2 * pnorm(-abs($\beta/ERR$)): $> PVAL$ to obtain the pvalues

* Step 1.1: Model fitted using dsBaseClient::ds.glm, based on the conventional iterative reweighted least squares (IRLS) algorithm [56], a variant of Newton optimization.

The data that is being shared with the client is always a result of an aggregation function (i.e. column sums and different products). For this reason, we added disclosure controls that guarantee that the aggregated data have a minimum number of valid points to prevent data leaks (as returning aggregates of a single individual will leak their data). The sacrifice we make with this approach is a slight imprecision in the results (quantified in the "Results" use case 1 section). This sacrifice is in exchange for computational performance, illustrated in Fig 7. It must be noted that performance is limited by the slowest server, therefore increasing the amount of data providers in the study does not imply increasing computational time as all servers work in parallel. Even though it is not explicitly mentioned on the pseudo-algorithm, differential privacy is also implemented on this algorithm and applied to all the operations that return aggregated results to the client. In this case it is trivial to see that there is no need to perform a sensitivity analysis, given the genetic data is encoded as [0, 1, 2] integer values.

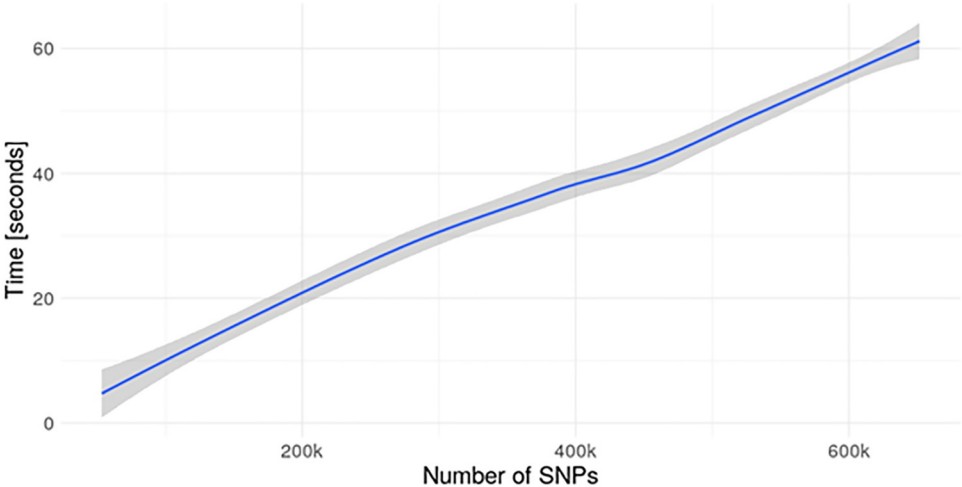

**Fig 7. Performance of the fast GWAS function.** The benchmark has been performed using a dataset consisting of 2,504 individuals and increasing SNPs to obtain the different data points. When computing this plot, Differential privacy (DP) was turned off. There are a total of 21 data points with 20 repetitions each one, the results are displayed using a smoothed "loess" line. The trend appears to be linear as more SNPs are added to the analysis, with an approximate rate of computation: 1M SNPs each 90 seconds. The description of the machine used for this benchmark is available on the "Methods" section. The results are subject to great fluctuation depending on the hard drive technology of the machine. Using state-of-the-art solid state drives the performance could be much better.

## Meta-level quality control

For researchers interested in performing a meta-level GWAS, it is possible to do this within OmicSHIELD. To accomplish this using established quality control protocols [37], three different plots are made available to assess the quality of the study level data:

- EAF Plot: Pinpoint harmonization problems.

- P-Z Plot: Issues with beta estimation. Can pinpoint errors in the pre-processing of the data.

- SE-N Plot: Can pinpoint issues with trait transformations.

These methods should provide researchers with enough information to assess whether values at the study level can be trusted. Any problems that arise from quality control should be notified to the data owners as OmicSHIELD does not provide tools for overcoming them as each case is likely to require an ad-hoc solution.

## Polygenic Risk Scores

The PRS is a statistic that is computed individually, therefore there are no concerns with having data distributed among multiple cohorts as results from individuals have no effect on the other cohorts The PRS is computed as:

$$PRS_i = \sum_{j=0}^{n} X_{ij}\beta_j$$

Where the genotype is $X$ (i is the individual index, j is the risk SNP index) and $\beta_j$ is the associated risk to the SNP j.

Given that PRS results are individual and not aggregated, they are disclosive. For this reason, the client does not receive any results when performing a PRS, results are instead stored

on local study servers. where they can be employed as covariates in subsequent analyses such as GWAS to control for genetic predisposition when identifying novel genetic associations, or utilized within GLM to explore interactions between genetic risk and environmental or lifestyle factors [57].

We can provide information on the SNPs and their beta-values from a given polygenic risk score by using data from the public repository PGS Catalog [40], which contains over 2,000 different literature-curated scores; a harmonization step ensures the risk alleles on the private study servers and the PGS Catalog are aligned.

### Baseline computations with GWASTools and SNPRelate

To assess the results yielded by our implementations, we used tools freely available to researchers: GWASTools (1.35.2), GENESIS (2.20.1) and SNPRelate (1.25.2).

### Experimental settings

We implemented our solutions on top of R 4.1.1. To evaluate OmicSHIELD, we emulated a network of data repository servers using DSLite 1.2.0. The solution was deployed using a Linux machine with two Intel Xeon Gold 6240 CPUs running at 2.6 GHz with 72 threads running on 36 cores and 256 GB of RAM. The hard drives are SAS drives at 10000 rpm 12Gb/s. Machine resources were equally distributed among the virtual data repositories.

### Acknowledgments

We acknowledge Sofia Aguilar on her valuable inputs on performing the methylation studies, dealing with methylation data and her efforts towards assessing the usability of OmicSHIELD on real-world data. We also thank Manuel Huth for his invaluable assistance in discussing differential privacy concepts and providing insightful feedback on the methods described in this paper.

### Author Contributions

**Conceptualization:** Juan R. Gonzalez.

**Formal analysis:** Xavier Escriba-Montagut.

**Funding acquisition:** Andrei S. Morgan, Rebecca C. Wilson, Paul Burton, Juan R. Gonzalez.

**Software:** Xavier Escriba-Montagut, Yannick Marcon, Augusto Anguita-Ruiz, Demetris Avraam, Jose Urquiza, Juan R. Gonzalez.

**Supervision:** Juan R. Gonzalez.

**Writing – original draft:** Xavier Escriba-Montagut, Yannick Marcon, Demetris Avraam, Juan R. Gonzalez.

**Writing – review & editing:** Xavier Escriba-Montagut, Juan R. Gonzalez.

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
