## [Decision Letter · Decision Letter 0]

18 Dec 2023

Dear Dr Gonzalez,

Thank you very much for submitting your manuscript "Federated privacy-protected meta- and mega-omic data analysis in multi-centre studies with a fully open source analytic platform" for consideration at PLOS Computational Biology.

As with all papers reviewed by the journal, your manuscript was reviewed by members of the editorial board and by several independent reviewers. In light of the reviews (below this email), we would like to invite the resubmission of a significantly-revised version that takes into account the reviewers' comments.

We cannot make any decision about publication until we have seen the revised manuscript and your response to the reviewers' comments. Your revised manuscript is also likely to be sent to reviewers for further evaluation.

Sincerely,

Gamze Gursoy

Guest Editor

PLOS Computational Biology

William Noble

Section Editor

PLOS Computational Biology

Reviewer's Responses to Questions

**Comments to the Authors:**

Reviewer #1: In this manuscript, the authors have developed OmicSHIELD a tool to support federated privacy-protecting analysis for different types of omics data including genomics, epigenomics and transcriptomics. This tool is open-source and builds on the existing DataSHIELD platform, which has been proposed to enable secure federated analysis of sensitive data.

Pros:

- OmicSHIELD is open-source, thus potentially facilitating reproducibility and integration with other downstream applications.

- Several privacy-protecting methods are considered in the tool, which can provide enhanced privacy protection for federated analysis, including disclosure traps, information filtering, and differential privacy.

- A variety of use-cases and omics analysis are supported by the tool.

Cons:

- Lack of guidelines on the use of privacy-protecting methods in practice.

- Limited novelty

Detailed Comments:

While the manuscript makes important step toward enabling federated multi-omics privacy-protecting analysis, there are some limitations that need to be addressed by the authors.

1. The proposed tool seems to support basic multi-omic analyses in federated settings. Recent advances in AI have led to numerous methods that leverage deep learning technology to improve the predictive outcomes in a variety of health applications (e.g., deep survival analysis, cancer subtyping). However, it is not clear whether the proposed tool can support these emerging predictive analyses.

2. Aspects regarding data partitioning across sites are not discussed with adequate details. In one use-case (PCA), the data are assumed to be horizontally partitioned. However, practical application settings may exhibit sites with vertically portioned data. Therefore, it is not clear, whether the proposed solution can be applied in those settings.

3. While OmicSHIELD enhances the privacy protection of federated analysis with the combination of multiple privacy-protecting technologies (e.g., differential privacy), the authors provided limited information on the use of such techniques in practice. As a result, users are left alone in finding the right balance between privacy and usability. It would be beneficial to add a use-case demonstrating these privacy techniques, including their impact on the usefulness and privacy, which may provide users with useful guidelines.

4. Minor comments: There are some missing spaces in some sentences (e.g., abstract and introduction). Also, it seems some citations to DataSHIELD are missing (e.g., introduction).

Reviewer #2: Gonzalez et al. present OmicSHIELD, a federated data analysis portal. OmicSHIELD is an R statistical language based implementation that uses DATAShield framework.

I found the study to be important and timely because genetic data privacy is one of the most important topics. The main contribution of the study is implementation of numerous genetic analysis methods using DataSHIELD, an existing library for privacy preserving analysis.

I have several other reservations regarding: Lack of description of contributions, not enough comparisons with existing methods; lack of how privacy protections are implemented and parametrized for DP; lack of more use cases; lack of clarity in the formulations.

1- Clarify the adversarial model and assumptions of the model with limitations of how privacy can be compromised, e.g., how does collusions between clients and data processors impact privacy? What can malicious adversaries do to breach privacy?

2- Following up from the previous comment, is OmicSHIELD compliant with GDPR? This should be justified clearly.

3- The manuscript misses numerous previous references about federated and privacy preserving/secure methods related to genomic data processing.

4- How does OmicSHIELD compare to similar existing platforms in regards to adversarial assumptions and functionalities?

5- I am not exactly sure about this statement: "However, pooled analysis in a multi-cohort setting is not recommended when data are heterogeneous among cohorts". Does this mean mega-analysis is not recommended when there is heterogeneity among populations? Can authors provide an example for a mega-analysis would fail in such as case?

6- In Figure 3, what are the non-disclosive statistics? Unless these are encrypted, it is not clear how a statistic can be non-disclosive.

7- It is very hard to follow the formulations without description of the symbols (e.g. page 18) and lack of appropriate mathematical notations which are currently rendered as text.

8- In figure 4, it is mentioned in one place that epsilon=3 is used for protection. This privacy budget is not justified since epsilon=3 is a fairly relaxed privacy budget. It is necessary to decrease the budget and compare results.

9- Following from previous comment, it is not clear how DP is implemented into OmicSHIELD. I think OmicSHIELD uses DataSHIELD's functions to impose DP but it is necessary to clarify what type of DP is implemented into OmicSHIELD with all fine grained details so that users can understand how privacy is preserved. Generally, DP-based methods do not perform well on large dimensional datasets.

10- Following from previous comment, \\Delta f is estimated via simulations (Page 18), which means it is not theoreticaly derived and thus it does not lead to exact DP.

11- Following from previous comment, \\Delta f calculation should be clarified for all data that are exchanged in all protocols.

12- Figure 5 shows a map of how data is distributed for expression/methylation analysis. Does that mean OmicSHIELD was run on multiple centers to perform a pooled analysis?

13- The privacy budget for DP is not reported for expression/methylation use case. Authors should show how the budget changes the accuracy of the final results.

14- There were no use cases for PCA, EWAS, and PRS.

15- In method descriptions, the "Data Processors" and "Clients" seems to be different entities but they were not clearly mentioned in the main text. I think DP_i refers to a data processor at the local site where data is stored. The setup of analysis should be clarified at the beginning of the study.

16- The details of PCA algorithm is not clear about how and why it would provide the same results as a pooled PCA analysis. DP_i sends u_i d_i to the client and client merges these and performs a final PCA. Do we expect that this will be the same as PCA on pooled data? Please provide more justification for this algorithm.

17- In PCA algorithm, sites need access to p_j, which must also be calculated in a privacy-preserving manner. It is not clear how this is accomplished.

18- Figure 6: What was the federated settings used in this plot?

19- Following up from previous comment, the authors should provide the time/network usage with a realistic analysis scenario. Currently I could not find any reporting of network requirements of the methods.

20- For PRS analysis authors state: "For this reason,

the client does not receive any results when performing a PRS, results are instead stored on local

study servers to be used as covariates in a GWAS or other aggregating functions."

Please provide some examples of how PRS estimates can be used as covariates in a GWAS or other aggregating functions.

Reviewer #3: The authors introduce, OmicSHIELD which is an open-source tool for federated analysis of sensitive omic data for biomedical research between multiple centers. I have two major concerns:

1. The paper introduces an open-source tool which is very interesting and useful for the community. Yet, the methods described in the paper lack technical novelty.

2. The paper itself, including the title, claim the system to be secure and private. Security and privacy are two different concepts that should be made clear in the respective subsections. Federated learning/analysis itself is not privacy-preserving due to several attacks. The authors state the system is privacy-preserving via its disclosure traps, control of output, and differential privacy. Also, authors state "the activity of authorized researchers can be audited and checks made to assess whether inference attacks have been performed" but there is no detailed information about how these audits are made. Eliminating inference attacks through only auditins is very challenging.

Without further details, it is very hard to quantify the privacy that the system provides (as also stated by the authors in Discussion with "but to personally assess and control the disclosure-prevention and differential-privacy mechanisms." If personal assesment is needed, this already underlines the system is not privacy-preserving by default. In the methods authors also state "Moreover, differential privacy can be applied to the results that the client receives."-> this should be mandatory for a system with aforementioned claims. I suggest authors to give a detailed explanation on all these issues.

- Regarding my previous comment, there is no experiment with differential privacy and different epsilon values. To actually protect privacy with differential privacy, the epsilon values should be reasonable and experimenting with different epsilon values would help the reader to understand the utility loss and the cutoff points.

- In introduction, authors state “To address this important issue, different privacy-protecting techniques such as federated learning (FL), differential privacy (DP), homomorphic encryption (HE), and secure multi-party computation (SMPC) have been developed, some or all of which may be adopted” -> federated learning alone is not a privacy-protecting technique as shown by most attack works. It should always be combined with a ‘privacy-preserving’ mechanism. Also, authors should add the relevant citations (works use DP,SMC, or HE) in this sentence.

- Algorithms developed by researchers could potentially be used alongside genotype-phenotype associations from genetic association studies by an attacker to predict genotypes and phenotypes of target individuals based on genome information shared by individuals or their relatives [11]. -> more relevant citation is needed for this claim such as the reconstuction attacks or kinship attack papers.

- ” It is developed for DataSHIELD (CITA DATASHIELD)” -> what is CITA DATASHIELD?

- “This framework has been used to develop FA in different settings such as machine learning and survival analysis (CITA dsMLP + dsSurvival)” -> what is CITA dsMLP + dsSurvival? It seems to me that the authors forgot citations.

- “This approach has been adopted as a countermeasure to inference attacks using complex queries [23]:” -> the authors should provide a proof for this claim and perform known attacks.

- “the differential privacy mechanisms we implement are intended to cause additional difficulty for such attacks. ” -> additional difficulty does not make a system secure and private as implied by the title.

- In Section “Omic analytic capabilities” authors state “OmicSHIELD contains functionalities to perform three types of omic data analysis: GWAS, DGE and EWAS” -> It is the first time DGE appears and authors should explain this acronym for the readers that are not familiar with the terminology

- There are other relevant works that authors should discuss:

1. Warnat-Herresthal, S., et al. Swarm learning for decentralized and confidential clinical machine learning. Nature 594, 265–270 (2021).

2. Sav, Sinem, et al. “Privacy-preserving federated neural network learning for disease-associated cell classification.” Patterns 3.5 (2022).

3. Li, W., et al. Privacy-preserving federated brain tumour segmentation. In International Workshop in Machine Learning in Medical Imaging (MLMI) (Springer) (2019).

4. Choudhury, O., et al. Differential privacy-enabled federated learning for sensitive health data(2019).

5. Jagadeesh, K.A., Wu, D.J., Birgmeier, J.A., Boneh, D., and Bejerano, G. (2017). Deriving genomic diagnoses without revealing patient genomes. Science 357, 692–695.

6. Cho, H., Wu, D., and Berger, B. (2018). Secure genome-wide association analysis using multiparty computation. Nat. Biotechnol. 36, 547–551.

Minor comments:

Title: multi-centre -> multi-center?

1.meta- and mega-analyses -> analysis. It is correctly written in some sections and there are typos in others

2. Bioconductor packages.OmicSHIELD -> space missing between sentences

3. “Therefore, our approach has the potential to fulfil the...” -> fulfill

4. Figure 1 caption: “dependences” -> dependencies, Figure 2 caption: “(e.g studies)” -> (e.g. studies), Figure 3 caption: “865 thousands. “-> 865 thousand.

5. Page 15: “...implements multiple generalized linear models separately (one per study) and combine the results...” -> combines

6. Page 15: “...described sexual dimorphism such is the case of...” -> such as

7. Discussion section: “multi-centre”-> multi-center?, multi center” -> multi-center, ” data sharing” -> data-sharing, “real life”-> real-life, “open source”-> open-source

8. "Computing principal components analyses (PCA) is the standard methodology to address populations in GWAS, but computing federated PCA is missed in other federated GWAS solutions (e.g. FAHME [8] or sPLINK [18])" -> is missing?

**Have the authors made all data and (if applicable) computational code underlying the findings in their manuscript fully available?**

Reviewer #1: None

Reviewer #2: Yes

Reviewer #3: Yes

PLOS authors have the option to publish the peer review history of their article (what does this mean?). If published, this will include your full peer review and any attached files.

Reviewer #1: No

Reviewer #2: No

Reviewer #3: No
---

## [Decision Letter · Decision Letter 1]

16 Apr 2024

Dear Dr Gonzalez,

Thank you very much for submitting your manuscript "Federated privacy-protected meta- and mega-omic data analysis in multi-center studies with a fully open source analytic platform" for consideration at PLOS Computational Biology.

As with all papers reviewed by the journal, your manuscript was reviewed by members of the editorial board and by several independent reviewers. In light of the reviews (below this email), we would like to invite the resubmission of a significantly-revised version that takes into account the reviewers' comments. Please note that the reviewer 3 was not able to review your revised manuscript and response. Therefore, we asked another reviewer for opinion, whose reviews are appended as Reviewer #4 below.

We cannot make any decision about publication until we have seen the revised manuscript and your response to the reviewers' comments. Your revised manuscript is also likely to be sent to reviewers for further evaluation.

Sincerely,

Gamze Gursoy

Guest Editor

PLOS Computational Biology

Sushmita Roy

Section Editor

PLOS Computational Biology

Reviewer's Responses to Questions

**Comments to the Authors:**

Reviewer #1: The revised manuscript improves the original submission. The addition of the differential privacy evaluations provides useful insights on the impact of the privacy parameter on the usability of the proposed approach. However, there are still some concepts about the privacy setting considered by the authors that need further clarification.

- It is still not clear how the proposed privacy techniques work in synergy to protect privacy. For example, the allele frequency estimation algorithm shows how differential privacy and thresholding are used, but those techniques are not directly deployed in PCA and GWAS analysis. The proposed framework should support diverse analytics with a consistent privacy protection. If different privacy mechanisms are used for each analysis, authors should justify the choice of the mechanism.

- Figure 5 is very helpful in illustrating the impact of different values of the privacy parameter on the usability. Yet, it is not clear how the sensitivity analysis is performed. One concern is that the privacy protection illustrated may hold at variant-level, resulting in a patient-level differential privacy protection that is much lower (e.g., multiple variants are correlated). What is the overall privacy guarantee for each individual? Additional details regarding the computation of the sensitivity and noise perturbation would help the reader.

- In the revision authors state that the proposed solutions should protect from “insiders with authorized access but malicious intent”. Please, provide further clarification about the adversarial model (e.g., honest-but-curios vs Byzantine).

- There are still some typos in the introduction.

Reviewer #2: Authors have clarified most of my comments sufficiently. I have two more comments, which can help make manuscript more clear before publication:

1) Authors added a citation for the sampling approach to clarify the comment on \\Delta f: "The assessment of Δ can be particularly complex for certain functions (e.g. limma + voom),

hence a sampling method [50] is used to assess Δ."

It would be very useful to the reader provide how this function is implemented step-by-step for the case of GWAS as an example.

Reviewer #4: The authors introduced an open-source software designed for the federated analysis of omics data, equipped with privacy protection techniques. The development of such a platform is useful as it facilitates data sharing without accessing raw data.

The privacy protection of the proposed software is mainly based on Differential Privacy. The client software may be used by a user who performs analyses frequently. Since the necessary privacy budget depends on the number of queries, it would be beneficial to present ideas on how to set the privacy budget and how to limit the number of queries in practice. The adversary model is described as malicious, but there is no explicit description in the manuscript why the omics data is protected from a malicious user. I think it may be better to describe why the system can protect data from malicious adversary. The security achieved by the system and potential security risks caused by major attacks are not clear. The software is composed of various packages, but their roles in ensuring security are not clear. For example, a more detailed explanation of OPAL and DATASHIELD would be helpful for the reader to understand the proposed software.

**Have the authors made all data and (if applicable) computational code underlying the findings in their manuscript fully available?**

Reviewer #1: None

Reviewer #2: Yes

Reviewer #4: Yes

PLOS authors have the option to publish the peer review history of their article (what does this mean?). If published, this will include your full peer review and any attached files.

Reviewer #1: No

Reviewer #2: No

Reviewer #4: No
---

## [Decision Letter · Decision Letter 2]

15 Jul 2024

Dear Dr Gonzalez,

Thank you very much for submitting your manuscript "Federated privacy-protected meta- and mega-omics data analysis in multi-center studies with a fully open source analytic platform" for consideration at PLOS Computational Biology. As with all papers reviewed by the journal, your manuscript was reviewed by members of the editorial board and by several independent reviewers. The reviewers appreciated the attention to an important topic. Based on the reviews, we are likely to accept this manuscript for publication, providing that you modify the manuscript according to the review recommendations.

The reviewers indicated that you have responded most of their concerns, however they also pointed out that there are still minor issues that need to be addressed in the differential privacy section as well as with the overstatement of the abilities of the tool. Before I can make a final decision, please address the remaining minor points in a revised manuscript. Please let me know if you have any questions.

Sincerely,

Gamze Gursoy

Guest Editor

PLOS Computational Biology

Sushmita Roy

Section Editor

PLOS Computational Biology

The reviewers indicated that you have responded most of their concerns, however they also pointed out that there are still minor issues that need to be addressed in the differential privacy section as well as with the overstatement of the abilities of the tool. Before I can make a final decision, please address the remaining minor points in a revised manuscript. Please let me know if you have any questions.

Reviewer's Responses to Questions

**Comments to the Authors:**

Reviewer #1: This revision addresses some of the previous issues. However, there are still a few aspects that in my opinion need to be clarified.

1. The inclusion of the pseudo-code for the resampling method originally proposed in [52] provides some useful information to the readers about the computation of the sensitivity. In my opinion, some improvement in the differential privacy section is needed. First, the approach in [52] provides an estimation of the sensitivity to achieve random differential privacy, which may result in a privacy protection that is weaker than the one provided by the traditional differential privacy model. Second, the number of rounds (N in the pseudo-code) cannot be arbitrary selected, but rather it needs to be carefully chosen to ensure that the privacy protection is met. The parameter N does not directly affect epsilon, instead a too small value may lead to failing the differential privacy guarantee overall. Note that [52] (Theorem 15) provides some lower bounds on the values of the parameters m and k in the original sensitivity sampler procedure, representing the number of samples and order statistics, respectively. Therefore, the statement “As the number of resamples performed can potentially impact differential privacy quality, we allow the data owner to configure this parameter”, may need to be adjusted to clarify the implications of these parameters. Also, pointers to the original paper indicating the privacy notion and information about the requirements on these parameters should be included.

2. There are still some typos and formatting issues. For example, the last three paragraphs in the introduction seem to be in a different font. Page 14: missing closing “)” after the link to Opal page. Page 21: “w’de added” -> “we have added”.

Reviewer #4: Minor comment:

The authors responded to most of my comments, and now I understand the adversarial model and the major risks they assumed.

Considering that the implemented privacy-preserving methods are not the type of methods that ensure no information is leaked (e.g., they guarantee only variant-level protection, have a query number limitation problem, and simple aggregation such as by disclosure traps cannot perfectly protect from re-identification attacks),

I think the manuscript includes rather over-advertisement descriptions such as:

- OmicSHIELD ensures that individual-level information remains confidential.

- It allows complex analyses across multiple datasets without compromising data security.

and it maybe better to mitigate the description such that the readers notice the limitations of the privacy-enhancing methods used in the proposed software.

**Have the authors made all data and (if applicable) computational code underlying the findings in their manuscript fully available?**

Reviewer #1: None

Reviewer #4: Yes

PLOS authors have the option to publish the peer review history of their article (what does this mean?). If published, this will include your full peer review and any attached files.

Reviewer #1: No

Reviewer #4: No

Figure Files:

Data Requirements:

Reproducibility:

References:

---

## [Editor Report · Decision Letter 3]

9 Sep 2024

Dear Dr Gonzalez,

I believe that the paper needs a little bit more clarity in privacy guarantees as I still share the concerns of reviewer, even after your edits to your statements.

If you can make it clear in the manuscript that the used mechanism (random differential privacy) may result in a privacy protection that is weaker than the one provided by the traditional

differential privacy model and also clarify the role of number of rounds in terms of how it needs to be carefully chosen and a very small value might fail the DP guarantees, I can quickly move forward with the decision.

Reviewer's comment was: "First, the

approach in [52] provides an estimation of the sensitivity to achieve random differential privacy,

which may result in a privacy protection that is weaker than the one provided by the traditional

differential privacy model. Second, the number of rounds (N in the pseudo-code) cannot be

arbitrary selected, but rather it needs to be carefully chosen to ensure that the privacy

protection is met. The parameter N does not directly affect epsilon, instead a too small value

may lead to failing the differential privacy guarantee overall. Note that [52] (Theorem 15)

provides some lower bounds on the values of the parameters m and k in the original sensitivity

sampler procedure, representing the number of samples and order statistics, respectively."

Best,

Gamze

Sincerely,

Gamze Gursoy

Guest Editor

PLOS Computational Biology

Sushmita Roy

Section Editor

PLOS Computational Biology

Dear authors,

I believe that the paper needs a little bit more clarity in privacy guarantees as I still share the concerns of reviewer, even after your edits to your statements.

If you can make it clear in the manuscript that the used mechanism (random differential privacy) may result in a privacy protection that is weaker than the one provided by the traditional

differential privacy model and also clarify the role of number of rounds in terms of how it needs to be carefully chosen and a very small value might fail the DP guarantees, I can quickly move forward with the decision.

Reviewer's comment was: "First, the

approach in [52] provides an estimation of the sensitivity to achieve random differential privacy,

which may result in a privacy protection that is weaker than the one provided by the traditional

differential privacy model. Second, the number of rounds (N in the pseudo-code) cannot be

arbitrary selected, but rather it needs to be carefully chosen to ensure that the privacy

protection is met. The parameter N does not directly affect epsilon, instead a too small value

may lead to failing the differential privacy guarantee overall. Note that [52] (Theorem 15)

provides some lower bounds on the values of the parameters m and k in the original sensitivity

sampler procedure, representing the number of samples and order statistics, respectively."

Best,

Gamze

Figure Files:

Data Requirements:

Reproducibility:

References:

---

## [Editor Report · Decision Letter 4]

10 Nov 2024

Dear Dr Gonzalez,

We are pleased to inform you that your manuscript 'Federated privacy-protected meta- and mega-omics data analysis in multi-center studies with a fully open source analytic platform' has been provisionally accepted for publication in PLOS Computational Biology.

Best regards,

Gamze Gursoy

Guest Editor

PLOS Computational Biology

Sushmita Roy

Section Editor

PLOS Computational Biology

Feilim Mac Gabhann

Editor-in-Chief

PLOS Computational Biology

Jason Papin

Editor-in-Chief

PLOS Computational Biology

---

## [Editor Report · Acceptance letter]

21 Nov 2024

PCOMPBIOL-D-23-01738R4 

Federated privacy-protected meta- and mega-omics data analysis in multi-center studies with a fully open source analytic platform

Dear Dr Gonzalez,

I am pleased to inform you that your manuscript has been formally accepted for publication in PLOS Computational Biology. Your manuscript is now with our production department and you will be notified of the publication date in due course.

With kind regards,

Lilla Horvath
